# Impacts of COVID-19 pandemic on professional identity development of intern nursing students in China: A scoping review

**Wen-ting Luo**, **Aimei Mao***

Kiang Wu Nursing College of Macau, Macau, Macau

* maoaimei@kwnc.edu.mo

## Abstract

### Purpose

Clinical experience plays a vital role in the development of the professional identity (PI) of nursing students. China has applied a strict zero- COVID health policy in combating the COVID-19 pandemic since December 2019 and studies have been conducted in different places of China to explore PI development of nursing students during the pandemic time among the intern nursing students who are on clinical practices. This review study aims to synthesize the previous studies and provide a comprehensive picture of the impacts of the pandemic on the PI development of intern nursing students.

### Methods

Arksey and O'Malley's five-stage scoping review framework was used. Combinations of keywords were used to search relevant articles in both Chinese and English databases published from inception of the articles until the final search date (10 March 2022). The initially included articles were also appraised for their quality, and those that passed the appraisal were left for data analysis. The analytic results were cross-checked among the reviewers.

### Result

Three themes emerged from the included studies: 1) the PI levels, 2) the impacts of personal and social factors of PL, and 3) the specific impact of the COVID-19 pandemic. The levels of students' PI ranged from 66%-80% of the total scores in PI instruments, almost the same levels as in pre-pandemic time, despite the elevated social image of nurses after the COVID-19 pandemic. There is no consensus about the impacts of most personal and social factors on students' PI across the studies. The impacts of COVID-19 on PI were both positive and negative.

### Conclusions

COVID-19 epidemic exerted complicated impacts on the PI of intern nursing students. While it is necessary to address the fear of the COVID-19 pandemic among intern nursing students, the pandemic may not be an opportunistic time to enhance the students' PI.

**Data Availability Statement:** All relevant data are within the paper and its Supporting Information files.

**Funding:** The author(s) received no specific funding for this work.

**Competing interests:** The authors have declared that no competing interests exist.

## Introduction

COVID-19 was first reported in December 2019 in central China and soon spread to other places of China and beyond the border. As of March 20, 2022, over 468 million confirmed cases and 6 million deaths were reported globally [1]. The COVID-19 pandemic has posed severe challenges for healthcare professionals, especially for the frontline nursing staff since its breakout in 2019 [1]. It also affected the clinical experience of nursing students [2, 3]. The pandemic is a fast-moving public health crisis and has manifested differently in different places [4]. While most countries have decided to live with COVID-19 after having experienced the painful surges of infections, China has kept the most stringent measures all the time. A number of research studies on nursing professional identity (PI) in China have explored the development of nursing students' PI in the context of the COVID-19 pandemic since early 2020 [5–9]. The current review study will synthesize the findings of the previous studies and portray a full picture of the relationship between the pandemic and PI of the intern nursing students.

Intern nursing students in China are those students who are in their final year of nursing program and are conducting one-year clinical practice under the supervision of experienced clinical nurses [10]. Nursing students are in the process of developing their PI, a journey when they internalize nursing-related knowledge, skills, and values [11, 12]. Junior nursing students develop PI in nursing schools by learning nursing-related knowledge and engaging with student peers and faculty members, while intern students nurture their PI by engaging with nursing staff and patients in clinical practices [13–15].

In many countries, COVID-19 has forced the shutdown of schools during the outbreak peaks [2, 16]. In addition, fieldwork practices for students have been canceled or postponed [2, 17]. China applies a zero-COVID policy, and even a handful of cases can trigger a big city with millions of population into lockdown. As a result, China has seen fewer and smaller COVID-19 spikes than many other countries. However, due to the strict policy and the high transmissibility of new COVID-19 variants, healthcare systems in many places in China have been overwhelmed by sporadic cases [18].

Intern nursing students have to face the changing learning environment brought by the COVID-19 pandemic. In some countries, like Spain, intern nursing students who had little work experience were put on the frontline in taking care of the COVID-19 cases, and this induced psychological distress among the intern students [19]. On the other hand, thanks to the relatively fewer COVID-19 inpatients in China than those in many other countries, no intern nursing students in China are reported to directly encounter COVID-19 patients because more experienced nurses or nurse specialists are usually allocated to the COVID-19 units. While frontline nurses exposed to COVID-19-related healthcare in China experienced COVID-19-induced distress [20–22], nursing students also reported anxiety, worry, fear, and sadness [22, 23].

Researchers interested in the PI of nursing students in China have added COVID-19 into their exploration of the influencing factors of PI since 2020. However, their studies differ in research areas and scopes, and the extent of the pandemic also varies in different places in China, resulting in different findings [5–9]. Therefore, it is difficult to obtain the convincing evidence from single studies on the specific role the pandemic has played in shaping the PI of nursing students. It is necessary to review current studies to get a broad view of the pandemic's impacts on the intern students' PI development.

## Methods

A scoping review of the literature on the PI of intern nursing students in China under the context of the COVID-19 pandemic was conducted to summarize the available evidence on the factors affecting the PI of intern nurses, with the review process guided by Arksey and

O'Malley's five-stage scoping review framework [24]. The researchers also referred to the Preferred Reporting Items for Systematic reviews and Meta-Analyses extension for Scoping Reviews (PRISMA-ScR) in data collection, data analysis, and the reporting of the review study [25] (Please see S1 Table).

### Stage 1: Identifying the research questions

Intern nursing students on clinical practices may have been differently affected by the pandemic compared with the junior students on the campus of the nursing schools. Two main questions were identified: 1) How were the PI levels of the intern nursing students during the epidemic? 2) What factors affected the PI of intern nursing students?

### Stage 2: Searching for relevant literature

The two researchers of this review study began with the Chinese databases CNKI (China National Knowledge Infrastructure) and Wanfang Data and then followed by English databases, including PubMed, CINAHL, Clinicalkey, and ProQuest. They also searched the grey literature on Google Scholar. No time limit and study design were imposed on the publications. The search was conducted to identify all relevant publications from inception until the final search date (10 March 2022).

### Stage 3: Screening articles

The inclusion and exclusion criteria for the included studies were developed. The studies would be included if they: 1) explored the impacts of COVID-19; 2) focused on the development of PI; and 3) contained intern nursing students from the baccalaureate programs. We targeted the students from the baccalaureate programs to reflect the nursing education trend in China. There are three levels of pre-registration nursing programs in China: secondary diploma; advanced diploma; and baccalaureate degree, with the baccalaureate program increasing steadily in the past years. Of around 540000 nursing students in the three programs in 2017 in China, 10% joined the baccalaureate program [26]; while among 515700 nursing students in 2012, 7.71% were in the baccalaureate program [27].

The studies would be excluded if they: 1) did not contain the nursing students in the baccalaureate program or 2) were conducted outside mainland China.

After the removal of duplicate articles, 1574 articles were left. A two-phase screening was independently performed by the two researchers. Phase 1: Review titles and abstracts of the articles according to inclusion and exclusion criteria. Phase 2: Read the full text of the articles left after the first phase. The two researchers independently screened articles, and any discrepancies between them were agreed upon through discussion. After the two screenings, 25 articles were identified (Fig 1).

**Quality appraisal.** As all the 25 articles were survey studies, the researchers conducted a quality appraisal using an appraisal checklist form (Center for Evidence-Based Management [28]) to examine the quality of the selected studies. This checklist includes 12 questions related to the quality of the articles, with three choices for each of the questions: Yes, No, or Can't tell. There had been no guidelines on how to use the appraisal checklist, and the researchers decided to quantify the appraisal outcome to facilitate the analysis. The desired answer to each question would be scored one mark. For Question 4 and Question 11, the answer "No" was scored one mark, while the answer "Yes" was scored one mark for the remaining ten questions. The reviewers independently assessed the articles and reached a consensus when there were dispensaries. The scores for the articles ranged from 6 to 9, with an average of 7.5, accounting for 62.5% of the total score. This means that, on average, the articles satisfied 62.5% of the

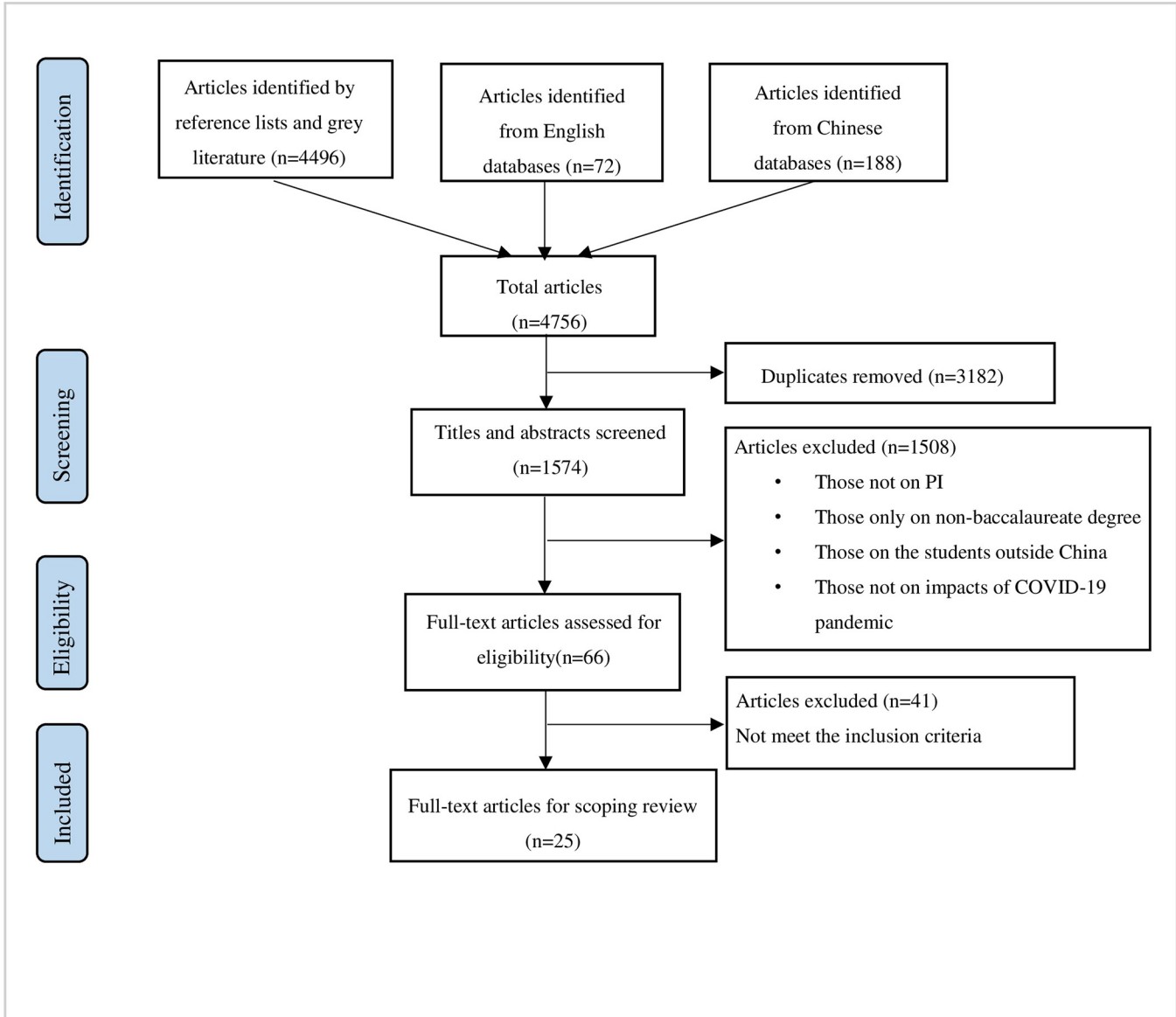

**Fig 1. The search and screening of relevant articles.**

quality requirements for survey studies. As all the 25 articles scored more than half of the total score, they were included in the article pool for reviewing (Please see S2 Table).

## Stage 4: Charting the data

According to the research purpose and questions, the 25 full-texted articles' essential information was extracted and classified. The basic information in the articles, such as authors, study titles, research design, participants, etc., was also collated. A chart was produced to collate the data.

## Phase 5: Synthesizing the findings

This process was done by applying the thematic analytic technique. The analysis mainly focused on influencing factors of PI development among nursing interns under the context of

the COVID-19 pandemic. Relevant data were compared, and relationships were detected. Again, the two researchers independently synthesized the findings in the included studies. The discrepancy was discussed until a consensus was reached.

## Results

### General information of the included studies

COVID-19 was first detected in December 2019 and then spread to other places. So it attracted wide attention from nursing professionals from 2020 on. The 25 articles published from 2020 to 2022 were quantitative survey studies. Fifteen studies were conducted exclusively among intern nursing students, and the others were among different groups of nursing students with the intern students as part of the samples. Table 1 represents the basic information of the included studies, and S3 Table provides detailed information on the studies' findings. Three themes emerged from the data of the included studies: the PI levels, the personal and social influencing factors of PI, and the specific impact of the COVID-19 pandemic.

### The PI levels

Six types of PI instruments were used in the included survey studies to examine the PI levels of the student participants, with higher scores indicating higher levels of the PI. The studies showed that the PI levels among nursing students ranged 67%-80% of the total scores. One study compared the PI levels of the intern students pre-and post-pandemic times and found that the PI levels post-pandemic were still at the over-medium levels despite a slight increase from pre-pandemic time [29]. Regarding the PI levels in different dimensions of the PI instruments, the social dimension scored higher than other dimensions, whereas the dimensions of job retention and autonomy in career choice scored lower.

### The personal and social influencing factors of the PI

Most of the included studies examined the relationships between nursing students' PI levels and their demographic backgrounds, such as gender, age, grade, etc. However, there are no confirmed findings across the studies. Some of the studies explored the impacts of participants' commitment to nursing on their PI levels, and these studies had common findings [8, 30–34]. For example, those who voluntarily chose nursing as their college major and/or those who intended to stay in nursing after graduation had higher levels of PI than those who were forced into nursing and/or who intended to leave nursing post-graduation [8, 30–34].

The included studies examined a variety of social factors, including healthcare-related social practices [7], the economic status of the family [34], and family members' support [32, 33]. While studies all highlighted the importance of healthcare-related practices in PI development, it was found that different social factors had different impacts [30, 35, 36].

### The specific impacts of COVID-19

The included studies implied that no intern students were assigned to take care of COVID-19 cases directly. Instead, the hospitals that hosted the intern students tended to assign experienced nursing staff to care for COVID-19 cases. However, the interns were still effected by the COVID-19 pandemic, and these effects could be positive or negative to the interns' PI development.

**The positive impacts.** During the COVID-19 pandemic, nurses played a key role in combating the pandemic and their heroic actions have been praised in mass media [37, 38]. This has been reflected in the self-perceptions of health professionals [39]. The review found that

**Table 1. Basic information of included studies (N = 25).**

| Author (year) | Title of study | Study design, sampling, data collection | Participants (Number) |
|---|---|---|---|
| Sun et al. (2021) | The exploration on levels and influencing factors of professional commitment among nursing undergraduates during COVID-19 epidemic | Cross-sectional survey. convenience sampling, online survey | Students including interns from baccalaureate program (N = 412) |
| Ma et al. (2020) | Analysis of occupational identity and influencing factors of internship nursing students under the COVID-19 | Cross-sectional survey, convenience sampling, online survey | Interns from baccalaureate program (N = 270) |
| Rao et al. (2021) | Analysis of the Status and Influencing Factors of Nursing Interns' Occupational Identity Under the NovelCoronavirus Pneumonia Epidemic | Cross-sectional survey, convenience sampling, online survey | Interns from Master's, baccalaureate, and advanced diploma programs (N = 208) |
| Yang et al. (2020) | Corona Virus Disease 2019 during the Period of Occupational Nursing Students' Occupation Identity and its Influencing Factors | Cross-sectional survey, convenience sampling, online survey | Interns from baccalaureate program (N = 227) |
| Yang et al. (2022) | Investigation on Professional Self-identity of Nursing Students during COVID-19 Epidemic | Cross-sectional survey, convenience sampling, online survey | Interns from advanced diploma and baccalaureate programs (N = 340) |
| Zhou et al. (2020) | The status and influencing factors of professional identity of nursing students during the period of COVID-19 | Cross-sectional survey, convenience sampling, online survey | Students including interns from advanced diploma and baccalaureate programs (N = 2000) |
| Hu et al. (2022) | Investigation on professional identity status of post-internship nursing students in Henan Province under the context of COVID-19 pandemic | Cross-sectional survey, convenience sampling, online survey | Interns from advanced diploma, baccalaureate, Master's programs (N = 625) |
| Luo et al. (2021) | Professional identity of nursing students in the context of COVID-19 outbreak: a cross-sectional study | Cross-sectional study, convenience sampling, online survey | Students including interns from baccalaureate program (N = 512) |
| Tang et al. (2020) | The influence of anxiety and coping style on professional identity of nursing students during internship in the context of COVID–19 | A cross-sectional survey, convenience sampling, online survey | Interns from secondary or advanced diploma (not clear) and baccalaureate programs (N = 230) |
| Shen et al. (2021) | The status and influencing factors of psychological capital and professional identity of undergraduate nursing students during COVID-19 | A cross-sectional survey, convenience sampling,online survey | Interns from baccalaureate program (N = 410) |
| Gao et al. (2020) | Current Situation of Nursing Interns' Perception of Stress and Its Impact on Professional Identity During COVID-19 | A cross-sectional survey, convenience sampling,online survey | Interns from secondary diploma, advanced diploma, and baccalaureate programs (N = 533) |
| Wen et al. (2021) | An investigation on the impact of the COVID-19 epidemic on the professional identity of undergraduate nursing students | A cross-sectional survey, random sampling,online survey | Interns from baccalaureate program (N = 260) |
| Liu et al. (2020) | A study on the impacts of COVID-19 epidemic on nursing professional identity | A cross-sectional survey, convenience sampling,online survey | Interns from secondary diploma, advanced diploma, baccalaureate, and master's programs (N = 587) |
| Zhang et al. (2021) | Professional identity and its relationship with emergency attitudes to public health emergencies among undergraduate nursing interns during the outbreak of COVID-19 | A cross-sectional survey, convenience sampling,online survey | Interns from baccalaureate program (N = 301) |
| Ruan (2021) | Analysis of professional identity and influencing factors of nursing students under public health emergencies | A cross-sectional survey, convenience sampling,online survey | Interns from baccalaureate program (N = 204) |
| Liu et al. (2020) | Analysis of the status and the influencing factos of nursing interns' work reshaping under COCID-19 context | A cross-sectional survey, convenience sampling, no descriptions on data collection | Interns from secondary diploma, advanced diploma, and baccalaureate programs (N = 287) |
| Li et al. (2022) | The status quo and correlation analysis of caring ability and professional identity of interns in the post pandemic era | A cross-sectional survey, convenience sampling,online survey | Interns from baccalaureate program (N = 293) |
| Huang et al. (2021) | Recognition of professional self-concept of nursing interns and its impact on professional attitudes under the COVID-19 epidemic | A cross-sectional survey, convenience sampling,online survey | Interns from baccalaureate program (N = 207) |
| Wang et al. (2021) | Research on professional identity and psychological resilience of college nursing students during prevention and control of COVID-19 epidemic | A cross-sectional survey, convenience sampling,online survey | Interns from baccalaureate program (N = 425) |
| Nie et al. (2021) | The Professional Identity of Nursing Students and Their Intention to Leave the Nursing Profession During the Coronavirus Disease (COVID-19) Pandemic | A cross-sectional survey, convenience sampling,online survey | Interns from advanced diploma and baccalaureate programs (N = 150) |
| Wang et al. (2020) | The effect of acute stress response on professional identity and self-efficacy of nursing students in China during COVID-19 outbreak: a cross-sectional study | A cross-sectional survey, convenience sampling,online survey | Students from baccalaureate program (N = 2024) |

*(Continued)*

**Table 1.** (Continued)

| Author (year) | Title of study | Study design, sampling, data collection | Participants (Number) |
|---|---|---|---|
| Zhang et al. (2021) | Professional identity of Chinese nursing students during the COVID-19 pandemic outbreak: A nation-wide cross-sectional study | A cross-sectional survey, convenience sampling,online survey | Students including interns from baccalaureate program (N = 6348) |
| Hao et al. (2020) | Investigation and research on the professional identity situation of nursing students under the background of COVID-19 epidemic | A cross-sectional survey,ramdom sampling,online survey | Students from secondary diploma, advanced diploma, and baccalaureate programs (N = 500) |
| Liu et al. (2021) | Current Situation Analysis of Nursing Students' Professional Attitudes and Employment Intentions during COVID-19 Pandemic | A cross-sectional survey, convenience sampling,online survey | Students from baccalaureate, Master, and Doctorate programs (N = 689) |
| Tang et al. (2021) | Associated factors of professional identity among nursing undergraduates during COVID-19: A cross-sectional study | A cross-sectional survey, convenience sampling,online survey | Students from baccalaureate program (N = 3875) |

the intern students' attitude towards nursing positively changed after the epidemic, as more students intended to stay in nursing after graduation [5, 29, 34, 40]. However, the students from the regions with COVID-19 outbreaks less agreed with the positive change than those from the areas without COVID-19 outbreaks [6, 41]. Some students expressed that the COVID-19 pandemic had made them more passionate about clinical nursing [5, 8, 9, 30, 40], and many of the intern students expressed willingness to join the frontline to fight COVID-19 if they were allowed to [7, 8, 42]. Those interns who did their clinical practice in designated hospitals where COVID-19 patients were treated scored higher levels of PI than other interns [34, 43].

**The negative impacts.**   The included studies also reported negative impacts on nursing students' PI by the COVID-19 pandemic. These impacts were mainly related to the psychological distress attributed to COVID-19. The studies reported that some nursing students experienced depression, anxiety, fear, and other mental health problems attributed to the COVID-19 pandemic and had lower PI levels. Although no studies examined the relationships between the students' knowledge about COVID-19 and the negative impacts of the pandemic on the psychological status of nursing students, some studies found that the students who had a higher level of knowledge of COVID-19 scored a higher level of PI [9, 44, 45].

Answers to the open questions in one study showed that the students worried about delayed or canceled clinical practicing opportunities, increased workload, and lack of health resources during the pandemic [34].

The influencing factors of the PI among intern nursing students under the COVID-19 pandemic can be illustrated in Fig 2.

## Discussion

This scoping review provided a summary of the status of PI and the influencing factors of the PI among intern nursing students in China during the COVID-19 pandemic. Compared with studies at the pre-pandemic time [46–48], this review added some interesting findings on PI development during a time when a public health crisis profoundly affected society.

While individual studies reported the influence of nursing students' personal factors on their PI, synthesis of the study findings did not show confirmed impacts for most factors. Previous reviews support that the demographic backgrounds of individual students can exert complicated influences on the students' PI status [46, 47]. This review study also maintained that the impacts of demographic factors of nursing students on their PI were contingency.

One notable change since the outbreak of the COVID-19 pandemic is the changed social attitudes towards nurses among the public. Since the COVID-19 cases were detected, nurses have risked their own lives in caring for patients and the public. They are hailed as heroes and

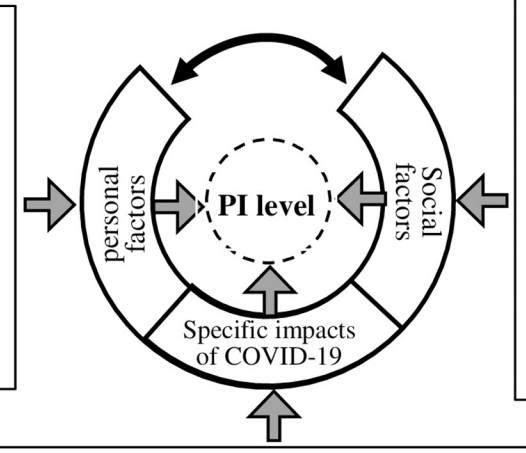

**Demographic factors**
- Grade
- Gender
- Educational Background
- School performance

**Commitment factors**
- Voluntarily chose nursing as their college major

**Social factors**
- Whether or not social practice is relevant to nursing
- Economic status of family
- Family members' support
- Family income
- Parents' job
- Parents were medical workers
- Clinical practices

**Specific impacts of COVID-19**
- The change of attitude toward Nursing
- The intent to which one knows about COVID-19 and its relevant knowledge and takes part in its training
- The social evaluation toward nurses
- The intensity and time of nursing workload
- The national reward for anti-epidemic personnel
- The willingness of the participation of frontline assistance
- Individual perceived stress by the epidemic
- Fixed point hospitals for COVID-19 and confirmed cases

**Fig 2. Factors effecting the PI of intern nursing.**

portrayed as hardworking and selfless professionals in mass media [37, 38]. While the elevated social image of nurses is observed in many countries [38, 49, 50], it is particularly important in the countries where the public do not value the social status of nurses [51, 52]. The changed social image of nurses was reflected in the higher levels of social dimensions of PI instruments, which is a startling contrast to the findings in the studies prior to the pandemic that social factors exerted an overall negative impact on PI development [47, 53, 54].

Surprisingly, the heightened levels in social dimensions of the PI instruments did not translate into increased overall PI levels among the nursing students. The scores of PI were 67%-80% of the total scores on the PI instruments. A pre-pandemic review on the PI status of nursing students reported that the PI levels of the students ranged 3–4 (60%-80%) to a total score of 5 on the PI instruments [47], indicating no apparent change of the PI levels pre- and post-pandemic times among the nursing students. The PI levels among the Chinese nursing students are the same as in other countries [51]. Further, the PI levels in the dimension of self-selection and preference for nursing on the PI instruments remained low both pre-and post-pandemic times, indicating that students were still forced into the nursing programs post-pandemic time, rather than self-selecting nursing as their college major [47, 51].

Another striking finding of our review study was that nursing students from regions with no COVID-19 cases had higher PI levels than those with the COVID-19 outbreaks. It is

supposed that when nursing students faced COVID-19 risks, they were more likely to be overwhelmed by the pandemic-triggered fear and anxiety rather than inspiration [55–57].

## Practical implications of the study findings

The unchanged PI levels among intern nursing students have implications for nursing education and nursing practice. Our review study implies that nursing schools in China had experienced difficulties recruiting students as they had to force the college candidates who had not chosen nursing into nursing. The recruitment difficulty is also reported in other countries where nurses have a low social image [58, 59]. Unfortunately, the recruitment difficulty may not be eased despite the increased social favor for nursing post-pandemic. The same passion for nursing during COVID-19 was once witnessed following the 2003 *Severe acute respiratory syndrome* outbreak [60, 61]. However, the passion triggered by Severe acute respiratory syndrome in 2003 quickly faded away and did not result in tangible impacts on the recruitment of nursing students and nurses. While nursing educators may capitalize on the COVID-19 pandemic to promote their students' reflection on nurses' PI, such reflection alone may not be enough to cultivate positive PI.

The fact that the PI levels among nursing students have not improved post-pandemic should be taken seriously by nursing professionals and policymakers. Some long-term problems nurses face, such as low pay, heavy workload, and low status within the healthcare power hierarchy, are amplified and exaggerated during the pandemic [62]. The International Council of Nurses [63] suggested that healthcare employers and organizations provide nurses with better working conditions and more support in the post-pandemic era. Politicians and policymakers should turn their appreciation for nurses, which was bred or enhanced during the pandemic, into positive changes, as did the Japanese Nurses Association, which, by lobbying the government, applied a series of strategies to support nurses and nursing students during the pandemic.

Our appraisal of the included studies revealed the generally low quality of the studies. Particularly, the weaknesses in sampling approaches and statistical analysis call for further training in research methodology in nursing schools and healthcare institutions for the current and future nursing researchers. We did not impose limitations in the search for relevant literature. However, the only type of the included studies as survey design indicates a lack of research diversity in the inquiry of nursing students' PI. One study included in the review contained open questions as part of a primarily quantitative inquiry and yielded limited but valuable data on the effects of COVID-19 on nursing students' PI development [34]. Research designs other than the survey, such as qualitative or experimental research, are needed in the future to explore nursing students' PI.

## Limitations of the study

Our review has two limitations: 1) The intern nursing students or stakeholders did not examine the review findings. According to Arksey and O'Malley [24], a consultation exercise should be conducted with the stakeholders so that the findings of scoping reviews can be practically helpful. The researchers of the review study are planning a qualitative study with intern nursing students, and they have decided to incorporate the findings of the review into their following study. 2) The researchers only included the studies that explicitly claimed to explore the PI of intern students and might have missed those that implicitly explored PI in nursing, such as self-concept, sense of career benefits, burnout, intention to leave nursing, etc.

## Conclusion

Our review study revealed some interesting findings on the impacts of the COVID-19 pandemic on the PI status of intern nursing students who are on clinical studies. The elevated

social image of nurses post-pandemic had influenced the self-perception of the intern students on nursing and nurses. However, the positive self-perception was not reflected in the students' PI levels. Negative impacts of the pandemic on students' PI were also observed, including psychological distress and disturbance in clinical study arrangements. China is still taking a zero-COVID health policy, and the impacts of the pandemic on intern students may continue. It may be unrealistic for nursing schools and healthcare institutions to use the positive effects of COVID-19 to elevate intern students' PI. Continued efforts from nursing professionals and policymakers are needed to address some long-them problems exacerbated by the COVID-19 pandemic.

## Supporting information

**S1 Table. PRISMA-ScR checklist.**
(DOCX)

**S2 Table. Quality appraisal of the articles.**
(DOCX)

**S3 Table. The results of included studies.**
(DOCX)

## Author Contributions

**Conceptualization:** Wen-ting Luo, Aimei Mao.

**Data curation:** Wen-ting Luo, Aimei Mao.

**Formal analysis:** Wen-ting Luo, Aimei Mao.

**Methodology:** Wen-ting Luo, Aimei Mao.

**Project administration:** Aimei Mao.

**Software:** Aimei Mao.

**Supervision:** Aimei Mao.

**Validation:** Aimei Mao.

**Visualization:** Wen-ting Luo, Aimei Mao.

**Writing – original draft:** Wen-ting Luo, Aimei Mao.

**Writing – review & editing:** Wen-ting Luo, Aimei Mao.

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
