## [Decision Letter · Decision Letter 0]

18 Aug 2022

PONE-D-22-18945Impacts of COVID-19 pandemic on professional identity development of intern nursing students in China: A scoping reviewPLOS ONE

Dear Dr. Luo,

Thank you for submitting your manuscript to PLOS ONE. After careful consideration, we feel that it has merit but does not fully meet PLOS ONE’s publication criteria as it currently stands. Therefore, we invite you to submit a revised version of the manuscript that addresses the points raised during the review process.

We look forward to receiving your revised manuscript.

Kind regards,

Jianguo Wang, PhD

Academic Editor

PLOS ONE

Journal Requirements:

Reviewers' comments:

Reviewer's Responses to Questions

**Comments to the Author**

1. Is the manuscript technically sound, and do the data support the conclusions?

Reviewer #1: Partly

2. Has the statistical analysis been performed appropriately and rigorously? 

Reviewer #1: N/A

3. Have the authors made all data underlying the findings in their manuscript fully available?

Reviewer #1: Yes

4. Is the manuscript presented in an intelligible fashion and written in standard English?

Reviewer #1: Yes

5. Review Comments to the Author

Reviewer #1: This is an interesting paper on the potential impact of COVID-19 on the professional identity of nursing students in China. Much discussion has been given to the role of the pandemic on the essential healthcare workforce, potentially contributing to burnout, anxiety, and perceptions of a lack of support from the general public. Overall the concept of the study is novel and contributes to the literature. The methods of the scoping review are sound and the approach is justified. The introduction and background is largely unsupported and many statements are made without citation or evidentiary support. It is confusing why there are two sections when a more traditional introduction section of a scientific paper would suffice. Regardless as to whether it is a unified or separate approach, however, there are too many statements made without citation. The results needs substantial revision as the presented findings do not fully reflect the objectives of the study.

Introduction

- Replace “has posed” to “pose” to avoid passive voice

- Line 107, replace “past two years” with the actual years, because if this paper is published and someone reads it 5 years from now, it is now inappropriate.

- Line 108, “It may also affect the …intern nursing students…” Is this a declaratory statement? It does not have a citation. If it is something you are trying to prove, then that is acceptable but should be listed in objectives instead. If it is just a declaratory statement, it needs citation or needs removal

- Line 108, similarly without citation and superfluous

- Citation needed for sentence ending on line 112

- Statements from line 113 – 117 reference studies but these are not cited.

- Please define what an intern nursing student is.

- Related to above, be clear that if this study uses international citations, whether the intern nursing student definition is context-specific

Background

- Why is there a background section and this not part of the introduction? Much of this seems to share similarity with the Introduction. If this is an editorial requirement of PLOS One then please ignore my comment. If it is not, these two sections (intro and Background) should be combined, trimmed, and revised for ease of reading.

- Page 7, line 134: “alternations” is a confusing word choice

- Page 7, lines 137-142. This is an uncited, un-referenced paragraph. The “up to down” governance in China may be familiar to the author but it will not be to any reader outside of China. Additionally, there is no evidentiary support given for these statements. Especially lines 141-142.

- Line 143-145, again, no citation for a statement that needs it

- Line 145-156, studies are mentioned but not referenced

Methods

- The first research question on line 162 is not clear. “How was the PI status?” is there a word missing?

- Please include the entire PubMed/MEDLINE search strategy, not broken into boxes as shown in Figure 1. The search must be able to be replicated exactly.

- Lines 181-190 need a rewrite for clarity and cohesion

-

Results

- As presented, Table 1 is illegible. Half of the first column is missing, as is the last column

- Table 2 needs some kind of massive revision. As presented, Table 2 is 19 pages long. The authors should consider a shorter, summary Table 2 and putting much of the rest of the information in the Appendix

- Related to the above comment about Table 2, the Results section needs revision. With a scoping review, much information is generated. It is unclear why the section on the instrumentation is needed, as it does not seem to provide much in regards to the selected research questions.

- Much like the issue with instrumentation, if the paper is focused on nursing PI in response to COVID-19, then the entire sections on personal and social factors is superfluous. It is not relevant to discuss gender, family, or interpersonal impacts on PI if these are completely unrelated to COVID-19, which they seem to be.

o Based off these large pieces of this review, it is recommended that the authors substantially revise the Results section. Table 1 and 2 need to be revised. As well, the authors have presented some interesting findings on the nurse PI/COVID dynamic, and this can (and should) be expanded further. By removing the unnecessary findings on non-COVID PI findings, the authors can then spend much more time synthesizing and summarizing the study findings on PI/COVID, which is the point of the paper.

- The results, as presented, do not match the theoretical framework for the qualitative approach described by the authors in Methods.

Discussion

- Line 359-363 are unsubstantiated

- The discussion on the instrumentation largely reflects the confusion and mis-placement of the instrumentation of Results. The study is on the impact of COVID-19 on nurse professional identity. It seems unsupported by the results to connect changes in this PI based on instrumentation, when the studies themselves were not reflective of COVID-19

- Line 436-440, “wishful thinking” is not a scientific statement

-

6. PLOS authors have the option to publish the peer review history of their article (what does this mean?). If published, this will include your full peer review and any attached files.

Reviewer #1: No

---

## [Author Response · Author response to Decision Letter 0]

29 Aug 2022

we are required a revision by the reviewer and we had made the suggested revision. The revised manuscript, together with responses to reviewer's comments, have been submitted. Please see the attachments. 

In addition, we checked our manuscript against the additional requirements from the editorial office.

---

## [Decision Letter · Decision Letter 1]

15 Sep 2022

Impacts of COVID-19 pandemic on professional identity development of intern nursing students in China: A scoping review

PONE-D-22-18945R1

Dear Dr. Luo,

We’re pleased to inform you that your manuscript has been judged scientifically suitable for publication and will be formally accepted for publication once it meets all outstanding technical requirements.

Kind regards,

Jianguo Wang, PhD

Academic Editor

PLOS ONE

Additional Editor Comments (optional):

Reviewers' comments:

Reviewer's Responses to Questions

**Comments to the Author**

1. If the authors have adequately addressed your comments raised in a previous round of review and you feel that this manuscript is now acceptable for publication, you may indicate that here to bypass the “Comments to the Author” section, enter your conflict of interest statement in the “Confidential to Editor” section, and submit your "Accept" recommendation.

Reviewer #1: All comments have been addressed

2. Is the manuscript technically sound, and do the data support the conclusions?

Reviewer #1: Yes

3. Has the statistical analysis been performed appropriately and rigorously? 

Reviewer #1: N/A

4. Have the authors made all data underlying the findings in their manuscript fully available?

Reviewer #1: Yes

5. Is the manuscript presented in an intelligible fashion and written in standard English?

Reviewer #1: Yes

6. Review Comments to the Author

Reviewer #1: The article is much improved. I still disagree with results focusing so heavily on non COVID PI factors, since the authors state definitively that it is the impact of COVID on PI that is their research focus. For such a considerable effort that went in to the scoping review, it is therefore surprising that only 1.5 double-spaced pages of the Results is dedicated to how COVID impacted nursing PI. I feel like this could be far more descriptive. But, that is left up to the editors to debate. The tables are also now far easier to read and wonderfully presented.

Some small editorial comments are listed for suggestion:

1. There are some issues with grammar in occasional places, but overall the writing is much improved

2. Some of the methods could be pushed to appendices, such as the full list of quality appraisal questions

Some needed corrections prior to publication:

1. Line 330 is missing in Discussion, so the sentence is not finished

2. Line 343 I am assuming the 67-80% PI score is from social dimension, given the previous sentence, but this should be reiterated. As written it can be confusing, as if PI scores are 80% of total PI score, which is nonsensical

3. Line 360 of discussion, this finding seemingly came completely out of the blue. How does the review of nurse PI imply that recruitment difficulties resulting in officials forcing students to enter nursing? Perhaps I missed this!

7. PLOS authors have the option to publish the peer review history of their article (what does this mean?). If published, this will include your full peer review and any attached files.

Reviewer #1: No

---

## [Editor Report · Acceptance letter]

29 Sep 2022

PONE-D-22-18945R1 

Impacts of COVID-19 pandemic on professional identity development of intern nursing students in China: A scoping review 

Dear Dr. Luo:

I'm pleased to inform you that your manuscript has been deemed suitable for publication in PLOS ONE. Congratulations! Your manuscript is now with our production department. 

Kind regards, 

on behalf of

Dr. Jianguo Wang 

Academic Editor

PLOS ONE